# EXPLORING AND ENHANCING THE TRANSFERABILITY OF ADVERSARIAL EXAMPLES

## ABSTRACT

State-of-the-art deep neural networks are vulnerable to adversarial examples, formed by applying small but malicious perturbations to the original inputs. Moreover, the perturbations can *transfer across models*: adversarial examples generated for a specific model will often mislead other unseen models. Consequently the adversary can leverage it to attack deployed systems without any query, which severely hinders the application of deep learning, especially in the safety-critical areas. In this work, we empirically study how two classes of factors those might influence the transferability of adversarial examples. One is about model-specific factors, including network architecture, model capacity and test accuracy. The other is the local smoothness of loss surface for constructing adversarial examples. Inspired by these understandings on the transferability of adversarial examples, we then propose a simple but effective strategy to enhance the transferability, whose effectiveness is confirmed by a variety of experiments on both CIFAR-10 and ImageNet datasets.

## 1 INTRODUCTION

Recently Szegedy et al. (2013) showed that an adversary is able to fool deep neural network models into producing incorrect predictions by manipulating the inputs maliciously. The corresponding manipulated samples are called *adversarial examples*. More severely, it was found that adversarial examples have cross-model generalization ability, i.e., that adversarial examples generated from one model can fool another different model with a high probability. We refer to such property as *transferability*. By now, more and more deep neural network models are applied in real-world applications, such as speech recognition, computer vision, etc.. Consequently, an adversary can employ the transferability to attack those deployed models without querying the systems (Papernot et al., 2016b; Liu et al., 2017), inducing serious security issues.

Understanding the mechanism of transferability could potentially provide various benefits. Firstly, for the already deployed deep neural network models in real systems, it could help to design better strategies to improve the robustness against the transfer-based attacks. Secondly, revealing the mystery behind the transferability of adversarial examples could also expand the existing understandings on deep learning, particularly the effects of model capacity (Madry et al., 2018; Fawzi et al., 2015) and model interpretability (Dong et al., 2017; Ross and Doshi-Velez, 2018). Therefore, studying the transferability of adversarial examples in the context of deep networks is of significant importance.

In this paper, we numerically investigate two classes of factors that might influence the adversarial transferability, and further provide a simple but rather effective strategy for enhancing the transferability. Our contributions are summarized as follows.

- We find that adversarial transfer is not symmetric, which means that adversarial examples generated from model $A$ can transfer to model $B$ easily does not implies the reverse is also natural. Second, we find that adversarial examples generated from a deep model appear less transferable than those from a shallow model. We also explore the impact of the non-smoothness of the loss surface. Specifically, we find that the local non-smoothness of loss surface harms the transferability of generated adversarial examples.

- Inspired by previous investigations, we propose the *smoothed gradient attack* to improve the adversarial transferability, which employs the locally averaged gradient instead of the

original one to generate adversarial examples. Extensive numerical validations justify the effectiveness of our method.

## 2 RELATED WORK

The phenomenon of adversarial transferability was first studied by Szegedy et al. (2013). By utilizing the transferability, Papernot et al. (2016b;a) proposed a practical black-box attack by training a substitute model with limited queried information. Liu et al. (2017) first studied the targeted transferability and introduced the ensemble-based attacks to improve the transferability. More recently, Dong et al. (2018) showed that the momentum can help to boost transferability significantly.

Meanwhile, there exist several works trying to explain the adversarial transferability. Papernot et al. (2016a) attributed the transferability to the similarity between input gradients of source and target models. Liu et al. (2017) proposed to use the visualization technique to see the large-scale similarity of decision boundaries. However, our numerical investigations imply that these similarity-based explanations have their intrinsic limitation that they cannot explain the non-symmetric property of adversarial transferability.

Our smoothed gradient attacks that enhance the transferability by utilizing the information in the small neighborhood of the clean example is inspired by the works on shattered gradients (Balduzzi et al., 2017) and model interpretability (Smilkov et al., 2017). Recently, similar strategies are also explored by Athalye et al. (2018a) and Warren He (2018) for white-box attacks, whereas we focus on the black-box settings. Our method is also related to the work by Athalye et al. (2018b), which introduced the expectation over transformation (EOT) method to increase robustness of adversarial examples. The EOT formulation is similar to our objective (7), but they did not study the transferability. Also our motivations are totally different from theirs.

## 3 BACKGROUND

### 3.1 ADVERSARIAL ATTACK

Let $f(x) : \mathbb{R}^d \mapsto \mathbb{R}^K$ denote the classifier. In deep learning, it is found that for almost any sample $x$ and its label $y^{\text{true}}$, there exists a small perturbation $\eta$ that is nearly imperceptible to human such that

$$\operatorname*{argmax}_i f_i(x) = y^{\text{true}}, \quad \operatorname*{argmax}_i f_i(x + \eta) \neq y^{\text{true}}. \tag{1}$$

We call $\eta$ the adversarial perturbation and correspondingly $x^{adv} := x + \eta$ the adversarial example. The attack (1) is called a non-targeted attack since the adversary has no control over which class the input $x$ will be misclassified to. In contrast, a *targeted attack* aims at fooling the model to produce a wrong label specified by the adversary, i.e. $\operatorname{argmax}_i f_i(x + \eta) = y^{\text{target}}$.

In this paper, we consider the pure black-box attacks. This means the adversary has no knowledge of the target model (e.g. architecture and parameters) and is also not allowed to query any input-output pair from the target model. On the contrary, the *white-box* attack means that the target model is available to the adversary.

### 3.2 GENERATING ADVERSARIAL EXAMPLES

**Modeling** In general, crafting adversarial examples can be modeled as the following optimization problem,

$$\begin{aligned} \text{maximize}_{x'} \quad & J(x') := J(f(x'), y^{\text{true}}) \\ \text{s.t.} \quad & \|x' - x\|_\infty \leq \varepsilon, \end{aligned} \tag{2}$$

where $J$ is some loss function measuring the discrepancy between the model prediction and ground truth; the $\ell_\infty$ norm $\|\cdot\|_\infty$ is used to quantify the magnitude of the perturbation. Other norm is also possible, but we focus on $\ell_\infty$ norm in this paper. To improve the strength of adversarial transferability, instead of using a single model, Liu et al. (2017) proposed the *ensemble attack*, which generates

adversarial examples against a model ensemble $f^1(x), f^2(x), \cdots, f^M(x)$:

$$\text{maximize}_{x'} \quad J\left(\frac{1}{M}\sum_{k=1}^{M} f^k(x'), y^{\text{true}}\right) \tag{3}$$
$$\text{s.t.} \quad \|x' - x\|_\infty \leq \varepsilon.$$

**Optimizer** We use the following iteration to solve (2) and (3),

$$x_{t+1} = \text{proj}_{\mathcal{D}}\left(x_t + \alpha\,\text{sign}(\nabla_x J(x_t))\right). \tag{4}$$

where $\mathcal{D} = [0, 255]^d \cap \{x' \mid \|x' - x\| \leq \varepsilon\}$ and $\alpha$ is the step size. We call the attack that evolves (4) for $T$ steps iterative gradient sign method (IGSM) attack (Kurakin et al., 2017; Madry et al., 2018). Furthermore, the famous fast gradient sign method (FGSM) is a special case with $\alpha = \varepsilon, T = 1$.

Dong et al. (2018) recently proposed the momentum-based attack as follows

$$g_{t+1} = \mu\, g_t + \nabla J(x_t)/\|\nabla J(x_t)\|_1$$
$$x_{t+1} = \text{proj}_D\left(x_t + \alpha\,\text{sign}(g_t)\right), \tag{5}$$

where $\mu$ is the decay factor of momentum. By using this method, they won the first-place in *NIPS 2017 Non-targeted Adversarial Attack and Targeted Adversarial Attack* competitions. We will call it mIGSM attack in this paper.

## 4 EVALUATION OF ADVERSARIAL TRANSFERABILITY

**Datasets** To evaluate the transferability, three datasets including MNIST, CIFAR-10 and ImageNet are considered. For ImageNet, evaluations on the whole ILSVRC2012 validation dataset are too time-consuming. Therefore, in each experiment we randomly select $5,000$ images that can be correctly recognized by all the examined models to form our new validation set.

**Models** (i) For MNIST, we trained fully connected networks (FNN) of $D$ hidden layers, with the width of each layer being 500. (ii) For CIFAR-10, we trained five models: *lenet,resnet20, resnet44, resnet56, densenet*. The test accuracies of them are $76.9\%, 92.4\%, 93.7\%, 93.8\%$ and $94.2\%$, respectively. (iii) For ImageNet, the pre-trained models provided by PyTorch are used. The Top-1 and Top-5 accuracies of them can be found on website[1]. To increase the reliability of experiments, all the models have been tested. However, for a specific experiment we only choose some of them to present since the findings are consistent among all the tested models.

**Criteria** Given a set of adversarial pairs, $\{(x_i^{adv}, y_i^{\text{true}})\}_{i=1}^N$, we calculate the *Top-1 success rate* (%) fooling a given model $f(x)$ by $\frac{100}{N}\sum_{k=1}^N \mathbb{1}[\text{argmax}_i f_i(x_k^{adv}) \neq y_k^{\text{true}}]$. For targeted attacks, each image $x^{adv}$ is associated with a pre-specified label $y^{\text{target}} \neq y^{\text{true}}$. Then we evaluate the performance of the targeted attack by the following Top-1 success rate: $\frac{100}{N}\sum_{k=1}^N \mathbb{1}[\text{argmax}_i f_i(x_k^{adv}) = y_k^{\text{target}}]$. The corresponding Top-5 rates can be computed in a similar way.

## 5 HOW MODEL-SPECIFIC FACTORS AFFECT TRANSFERABILITY

Previous studies on adversarial transferability mostly focused on the influence of attack methods (Liu et al., 2017; Dong et al., 2018; Tramèr et al., 2017; Kurakin et al., 2017). However it is not clear how the choice of source model affects the success rate transferring to target models. In this section, we investigate this issue from three aspects including architecture, test accuracy and model capacity.

### 5.1 ARCHITECTURE

We first explore how the architecture similarity between source and target model contributes to the transferability. This study is crucial since it can provide us guidance to choose the appropriate source models for effective attacks. To this end, three popular architectures including ResNet, DenseNet and VGGNet are considered, and for each architecture, two networks are selected. Both one-step and multi-step attacks are performed on ImageNet dataset. Table 1 presents the experiment results, and the choice of hyper-parameters is detailed in the caption.

---

[1]http://pytorch.org/docs/master/torchvision/models.html

Table 1: Top-1 success rates(%) of FGSM and IGSM attacks. The row and column denote the source and target models, respectively. For each cell, the left is the success rate of FGSM ($\varepsilon = 15$), while the right is that of IGSM ($T = 5, \alpha = 5, \varepsilon = 15$). The dashes correspond to the white-box cases, which are omitted.

| | resnet18 | resnet101 | vgg13_bn | vgg16_bn | densenet121 | densenet161 |
|---|---|---|---|---|---|---|
| resnet18 | - | 36.9 / 43.4 | 51.8 / 58.0 | 45.1 / 51.7 | 41.1 / 49.2 | 30.0 / 35.8 |
| resnet101 | 48.5 / 57.2 | - | 38.9 / 41.6 | 33.1 / 40.0 | 33.2 / 46.9 | 28.7 / 43.2 |
| vgg13_bn | 35.5 / 26.8 | 14.8 / 10.8 | - | 58.8 / 90.7 | 19.1 / 15.9 | 13.8 / 11.7 |
| vgg16_bn | 35.2 / 26.1 | 15.6 / 11.1 | 61.9 / 91.1 | - | 21.1 / 16.8 | 15.8 / 13.2 |
| densenet121 | 49.3 / 63.8 | 34.4 / 50.7 | 47.6 / 58.7 | 41.0 / 57.8 | - | 38.5 / 73.6 |
| densenet161 | 45.7 / 56.3 | 33.8 / 54.6 | 48.6 / 56.0 | 41.3 / 55.9 | 43.4 / 78.5 | - |

We can see that the transfers between two models are non-symmetric, and this phenomenon is more obvious for the models with different architectures. For instance, the success rates of IGSM attacks from *densenet121* to *vgg13_bn* is 58.7%, however the rate from *vgg13_bn* to *densenet121* has only 15.9%. The lack of symmetry implies that previous similarity-based explanations of adversarial transferability are quite limited.

Another interesting observation is that success rates between models with similar architectures are always much higher. For example the success rates of IGSM attacks between *vgg13_bn* and *vgg16_bn* are higher than 90%, nearly twice the ones of attacks from any other architectures.

## 5.2 Model Capacity and Test Accuracy

We first study this problem on ImageNet dataset. A variety of models are used as source models to perform both FGSM and IGSM attacks against *vgg19_bn*, and the results are displayed in Figure 1. The horizontal axis is the Top-1 test error, while the vertical axis is the number of model parameters that roughly quantifies the model capacity.

We can see that the models with powerful attack capability concentrate in the bottom left corner, whereas for those models with either large test error or large number of parameters, the fooling rates are much lower. We also tried other models with results shown in Appendix C , and the results show no difference.

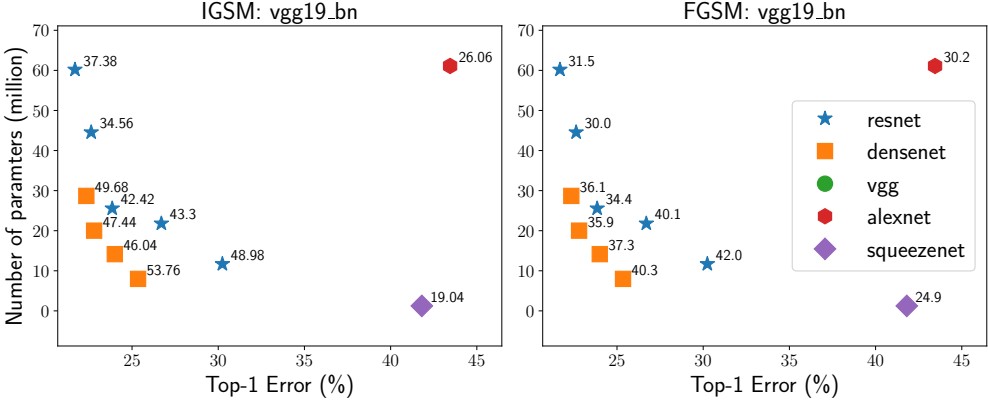

Figure 1: Top-1 success rates of IGSM ($T = 20, \alpha = 5, \varepsilon = 15$) and FGSM($\varepsilon = 15$) attacks against *vgg19_bn* for various models. The annotated value is the success rate transferring to the *vgg_19*. Here, the models of vgg-style have been removed to exclude the influence of architecture similarity. For the same color, the different points corresponds networks of different depths.

We suspect that the impact of test accuracy is due to that the decision boundaries of high-accuracy models are similar, since they all approximate the ground-truth decision boundary of true data very well. On the contrary, the model with low accuracy has a decision boundary relatively different from the high-accuracy models. Here the targeted network *vgg19_bn* is a high-accuracy model. Therefore it is not surprising to observe that high-accuracy models tend to exhibit stronger attack capability. This phenomenon implies a kind of data-dependent transferability, which is different from the architecture-specific transfers observed in the previous section.

It is somewhat strange that *adversarial examples generated from deeper model appear less transferable*. To further confirm this observation, we conduct additional experiments on MNIST and CIFAR-10. Table 2 shows the results, which is basically consistent. This observation suggests us not to use deep models as the source models when performing transfer-based attacks, although we have not fully understand this phenomenon.

Table 2: Top-1 success rates (%) of attack from the source model (row) to the target model (column). (a) FGSM attack for MNIST with $\varepsilon = 40$ and $D$ denotes the depth of the fully connected network. (b) FGSM attack for CIFAR-10 with $\varepsilon = 10$.

<table>
<tr><td colspan="5" align="center">(a) MNIST</td><td colspan="5" align="center">(b) CIFAR-10</td></tr>
<tr><td></td><td>$D=0$</td><td>$D=2$</td><td>$D=4$</td><td>$D=8$</td><td></td><td>resnet20</td><td>resnet44</td><td>resnet56</td><td>densenet</td></tr>
<tr><td>$D=0$</td><td>-</td><td>62.9</td><td>62.9</td><td>64.4</td><td>resnet20</td><td>-</td><td>70.4</td><td>64.0</td><td>71.6</td></tr>
<tr><td>$D=2$</td><td>52.9</td><td>-</td><td>48.3</td><td>49.4</td><td>resnet44</td><td>65.4</td><td>-</td><td>57.1</td><td>65.8</td></tr>
<tr><td>$D=4$</td><td>47.3</td><td>43.1</td><td>-</td><td>44.8</td><td>resnet56</td><td>66.2</td><td>62.9</td><td>-</td><td>40.3</td></tr>
<tr><td>$D=8$</td><td>31.2</td><td>29.2</td><td>29.0</td><td>-</td><td></td><td></td><td></td><td></td><td></td></tr>
</table>

# 6 NON-SMOOTHNESS OF THE LOSS SURFACE

In this section, we consider that how the smoothness of loss surface $J(x)$ affects the transferability. For simplicity, let $g(x) = \nabla_x J(x)$ denote the gradient. Smilkov et al. (2017) showed that gradient $g(x)$ is very noisy and uninformative for visualization, though the model is trained very well. Balduzzi et al. (2017) studied a similar phenomenon that gradients of deep networks are extremely shattered. Both of them implies that the landscape is locally extremely rough. We suspect that this local non-smoothness could harm the transferability of adversarial examples.

## 6.1 INTUITION

For simplicity, assume model A and B are the source and target models which are well trained with high test accuracies, respectively. Previous methods use $g_A(x)$ to generate adversarial perturbations, so the transferability mainly depends on how much sensitivity of $g_A$ for model A can transfer to model B. As illustrated in Figure 2, where three curves denote the level sets of three models, we can see that the non-smoothness can hurt the transferability, since both model A and B have very high test accuracy, their level sets should be similar globally, and $J_B(x)$ is probably unstable along $g_A$. As illustrated, the local oscillation of $g_A$ makes the sensitivity less transferable. One way to alleviate this is to smooth the landscape $J_A$, thereby yielding a more transferable gradient $G_A$, i.e. $\langle \hat{G}_A, \hat{g}_B \rangle > \langle \hat{g}_A, \hat{g}_B \rangle$. Here we use $G$ denote the gradient of the smoothed loss surface.

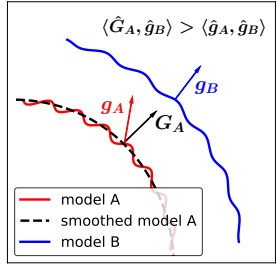

Figure 2: Illustration of how the non-smoothness of the loss surface hurts the transferability. For any $\boldsymbol{a}$, let $\hat{\boldsymbol{a}}$ denote the unit vector $\frac{\boldsymbol{a}}{\|\boldsymbol{a}\|_2}$.

## 6.2 JUSTIFICATION

To justify the previous arguments, we consider smoothing the loss surface by convolving it with a Gaussian filter, then the smoothed loss surface is given by $J_\sigma(x) := \mathbb{E}_{\xi \sim \mathcal{N}(0,I)}[J(x + \sigma\xi)]$. The corresponding gradient can be calculated by

$$G_\sigma(x) = \mathbb{E}_{\xi \sim \mathcal{N}(0,I)}[g(x + \sigma\xi)]. \tag{6}$$

The extent of smoothing is controlled by $\sigma$, and please refer to Appendix A to see the smoothing effect. We will show that $G_\sigma(x)$ is more transferable than $g(x)$ in the following.

**Gradient similarity** We first quantify the cosine similarity between gradients of source and target models, respectively. Two situations are considered: *vgg13_bn→vgg16_bn*, *densenet121→vgg13_bn*, which correspond the within-architecture and cross-architecture transfers, respectively. We choose $\sigma = 15$, and the expectation in (6) is estimated by using $\frac{1}{m}\sum_{i=1}^{m} g(x + \xi_i)$. To verify the averaged gradients do transfer better, we plot the cosine similarity against the number of samples $m$. In Figure 3a, as expected, we see that the cosine similarity between $G_A$ and $g_B$ are indeed larger than

the one between $g_A$ and $g_B$. Moreover, the similarity increases with $m$ monotonically, which further justifies that $G_A$ is more transferable than $g_A$.

**Visualization** In Figure 3b we visualize the transferability by comparing the decision boundaries of model A (*resnet34* ) and model B (*densenet121*). The horizontal axis represents the direction of $G_A$ of *resnet34*, estimated by $m = 1000, \sigma = 15$, and the vertical axis denotes orthogonal direction $h_A := g_A - \langle g_A, \hat{G}_A \rangle \hat{G}_A$. This process means that we decompose the gradient into two terms: $g_A = \alpha G_A + \beta h_A$ with $\langle G_A, h_A \rangle = 0$ . Each point in the 2-D plane corresponds to the image perturbed by $u$ and $v$ along each direction, clip$(x + u \hat{G}_A + v \hat{h}_A, 0, 255)$, where $x$ is the clean image. The color corresponds to the label predicted by the target model.

It can be easily observed that for model A, a small perturbation in both directions can produce wrong classification. However, when applied to model B, the sensitivities of two directions dramatically change. The direction of $h_A$ becomes extremely stable, whereas to some extent $G_A$ preserves the sensitivity, i.e. $G_A$ do transfer to model B better than $h_A$. This suggests that for gradient $g_A$, the noisy part $h_A$ is less transferable than the smooth part $G_A$. We also tried a variety of other models shown in Appendix C, and the results are the same.

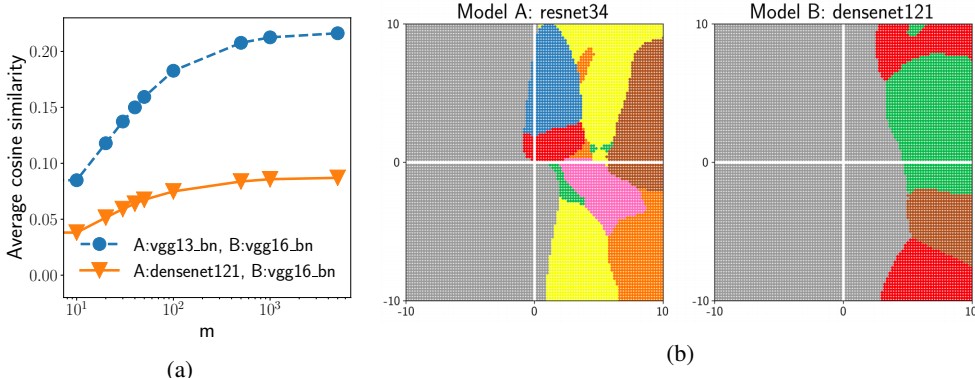

(a)

(b)

Figure 3: (a) Cosine similarity between the gradients of source and target models. (b) Visualization of decision boundaries. The origin corresponds to the clean image shown in Figure 10 of Appendix. The same color denotes the small label, and the gray color corresponds to the ground truth label.

# 7 SMOOTHED GRADIENT ATTACK

## 7.1 METHOD

Inspired by the previous investigations, enhancing the adversarial transferability can be achieved by smoothing the loss surface. Then our objective becomes,

$$\text{maximize} \quad J_\sigma(x') := \mathbb{E}_{\xi \sim \mathcal{N}(0, I)}[J(x' + \sigma\xi)]$$
$$\text{s.t.} \quad \|x' - x\| \leq \varepsilon. \tag{7}$$

Intuitively, this method can also be interpreted as generating adversarial examples that are robust to Gaussian perturbation. Expectedly, the generated robust adversarial examples can still survive easily in spite of the distinction between source and target model.

If use the iterative gradient sign method to solve (7), we have the following iteration:

$$G_t = \frac{1}{m} \sum_{i=1}^{m} \nabla J(x_t + \xi_i), \quad \xi_i \sim \mathcal{N}(0, \sigma^2 I)$$
$$x_{t+1} = \text{proj}_{\mathcal{D}} \left( x_t + \alpha \, \text{sign} \left( G_t \right) \right), \tag{8}$$

where $G_t$ is a mini-batch approximation of the smoothed gradient (6). Compared to the standard IGSM, the gradient is replaced by a smoothed version, which is endowed with stronger transferability. Therefore we call this method sg-IGSM. The special case $T = 1, \alpha = \varepsilon$, is accordingly called sg-FGSM. Any other optimizer can be used to solve the (7) as well, and we only need to replace the original gradient with the smoothed one.

## 7.2 CHOICE OF HYPER PARAMETERS

We first explore the sensitivity of hyper parameters $m, \sigma$ when applying our smoothed gradient technique. We take ImageNet dataset as the testbed, and sg-FGSM attack is examined. To increase the reliability, four attacks are considered here. The results are shown in Figure 4.

We see that sg-FGSM consistently outperforms FGSM for all distortion level, although the improvement varies for different $\varepsilon$. Furthermore, larger $m$ leads to higher success rate due to the better estimation of the smooth part of gradient, and the benefit starts to saturate after $m \geq 30$. For the smoothing factor $\sigma$, we find neither too large nor too small value can work well, and the optimal $\sigma$ is about 15. Overly large $\sigma$ will introduce a large bias in (8), and extremely small $\sigma$ is unable to smooth the landscape enough.

Therefore, in the subsequent sections, we will use $m = 20, \sigma = 15$ to estimate the smoothed gradient and only report the result for one distortion level.

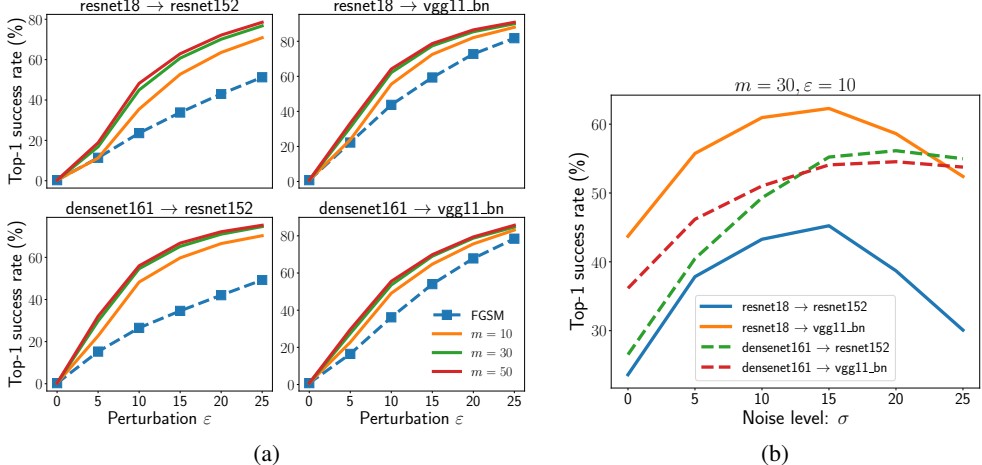

Figure 4: **(a)** Success rates (%) for sg-FGSM attacks with different $m$. Here we use $\sigma = 15$. **(b)** The sensitivity of the hyper parameter $\sigma$.

## 7.3 EFFECTIVENESS

**Single-model attack**    We first test the effectiveness of our method for single-model based attacks on non-targeted setting. To make a fair comparison, we fix the number of gradient calculation per sample at 100. Specifically, for sg-IGSM we have $T = 5$ due to $m = 20$, whereas $T = 100$ for IGSM. The results are shown in Table 3. We see that smoothed gradients do enhance the transferability dramatically for all the attacks considered. Please especially note those bold rates, where the improvements have reached about 30%.

Table 3: Top-1 success rates(%) of non-targeted IGSM and sg-IGSM attacks. The row and column denote the source and target modesl, respectively. For each cell, the left is the success rate of IGSM ($T = 100, \alpha = 1$), while the right is the that of sg-IGSM ($T = 5, \alpha = 5$). In this experiment, distortion level $\varepsilon = 15$.

|  | densenet121 | resnet152 | resnet34 | vgg13_bn | vgg19_bn |
|---|---|---|---|---|---|
| densenet121 | - | **50.1 / 80.6** | 59.9 / 87.2 | 62.2 / 82.2 | 56.5 /84.3 |
| resnet152 | **52.5 / 81.3** | - | 57.2 / 85.6 | 47.7 / 71.1 | **42.9 / 72.6** |
| resnet34 | 51.5 / 76.4 | 46.5 / 73.1 | - | 53.8 / 74.8 | 49.1 / 74.5 |
| vgg13_bn | 24.1 / 49.2 | 14.3 / 33.5 | **25.1 / 54.1** | - | 90.6 / 96.4 |
| vgg19_bn | **27.1 / 57.5** | 16.7 / 41.6 | **27.6 / 60.7** | 92.0 / 96.1 | - |

**Ensemble attack**    In this part, we examine the ensemble-based attack on the targeted setting[2]. For targeted attacks, generating an adversarial example predicted by target models as a specific label is

---

[2]Compared to non-targeted attack, we find that a larger step size $\alpha$ is necessary for generating strong targeted adversarial examples. Readers can refer to Appendix B for more details on this issue, though we cannot fully understand it. Therefore a much larger step size than the non-target attacks is used in this experiment.

too hard, resulting in a very low success rate. We instead adopt both Top-1 and Top-5 success rates as our criteria to better reflect the improvement of our method. A variety of model ensembles are examined, and the results are reported in Table 4.

Table 4: Top-5 success rates (%) of ensemble-based approaches for IGSM ($T = 20, \alpha = 15$) versus sg-IGSM ($T = 20, \alpha = 15$). The row and column denote the source and target models, respectively. The Top-1 rates can be found in Appendix C, which show the same trend as the Top-5 rates.

| Ensemble | resnet152 | resnet50 | vgg16_bn |
|---|---|---|---|
| resnet101+densenet121 | 28.1 / 56.8 | 26.2 / 52.4 | 8.1 / 29.7 |
| resnet18+resnet34+resnet101+densenet121 | **50.4 / 70.4** | 54.7 / 72.4 | 28.1 / 52.6 |
| vgg11_bn+vgg13_bn+resnet18 +resnet34+densenet121 | **24.3 / 55.8** | **36.9 / 65.9** | 62.2 / 83.5 |

As we can see, it is clear that smoothed gradient attacks outperform the corresponding normal ones by a remarkable large margin. More importantly, the improvement never be harmed compared to single-model case in Table 3, which implies that *smoothed gradient can be effectively combined with ensemble method without compromise*.

**Momentum attack** In this experiment, three networks of different architectures are selected. As suggested in Dong et al. (2018), we choose $\mu = 1$, and all attacks are iterated for $T = 5$ with step size $\alpha = 5$. In Table 5, we report the results of non-targeted attacks of three attacks including mIGSM, sg-IGSM and mIGSM with smoothed gradient (sg-mIGSM). It is shown that our method clearly outperforms momentum-based method for all the cases. Moreover, by combining with the smoothed gradient, the effectiveness of momentum attacks can be further improved significantly.

Table 5: Top-1 success rates (%) of momentum attacks and smoothed gradient attacks. The row and column denote the source and target model, respectively. Each cell contains three rates corresponding to mIGSM, sg-IGSM and sg-mIGSM attacks, respectively.

| | resnet18 | densenet121 | vgg13_bn |
|---|---|---|---|
| resnet18 | - | 65.6 / 73.1 / 86.5 | 70.4 / 77.7 / 86.7 |
| densenet121 | 72.7 / 84.5 / 91.1 | - | 68.7 / 80.3 / 86.7 |
| vgg13_bn | 43.1 / 58.6 / 74.3 | 28.4 / 44.7 / 60.9 | - |

## 7.4 ROBUSTNESS

Smoothed gradient attacks can be viewed as generating adversarial examples robust against Gaussian noise perturbations. Therefore, we are interested in how robust is the adversarial example against other image transformations. To quantify the influence of transformations, we use the notion of destruction rate defined by Kurakin et al. (2017). The lower is this rate, the more robust are the adversarial examples.

*Densenet121* and *resnet34* are chosen as our source and target model, respectively; and four image transformations are considered: rotation, Gaussian noise, Gaussian blur and JPEG compression. Figure 5 displays the results, which show that adversarial examples generated by our methods are more robust than those generated by vanilla methods. This numerical result is interesting, since we only explicitly increase the robustness against Gaussian noise in generating adversarial examples. We speculate that the robustness can also transfer among different image transforms.

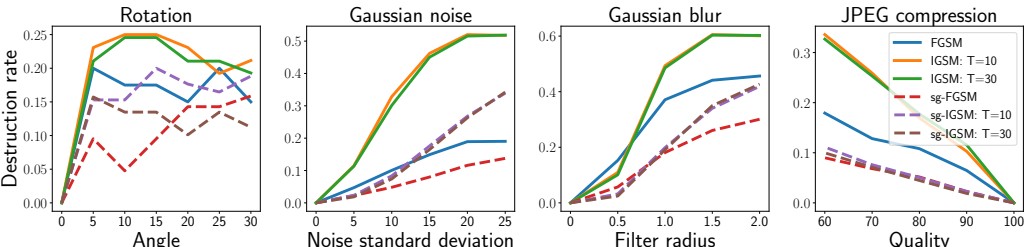

Figure 5: Destruction rates of adversarial examples for various methods. For smoothed gradient attacks, we choose $m = 20, \sigma = 15$. The distortion $\varepsilon = 15$.

# 8 CONCLUSION

In this paper, we first investigated the influence of model-specific factors on the adversarial transferability. It is found that the model architecture similarity plays a crucial role. Moreover models with lower capacity and higher test accuracy are endowed with stronger capability for transfer-based attacks. we second demonstrate that the non-smoothness of loss surface hinders the transfer of adversarial examples. Motivated by these understandings, we proposed the smoothed gradient attack that can enhance the transferability of adversarial examples dramatically. Furthermore, the smoothed gradient can be combined with both ensemble and momentum based approaches rather effectively.

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

## A  VISUALIZATION OF SMOOTHED GRADIENT

In this section, we provide an visualization understanding how the local average has the smoothing effect. We choose *densenet121* as the model and visualize the saliency map of gradient $\nabla_x J(x)$ and the smoothed versions for varying $m$. Two images are considered, and the results are shown in Figure 6. We can see that local average can smooth the gradient significantly. Please refer to the work by Smilkov et al. (2017) for more details.

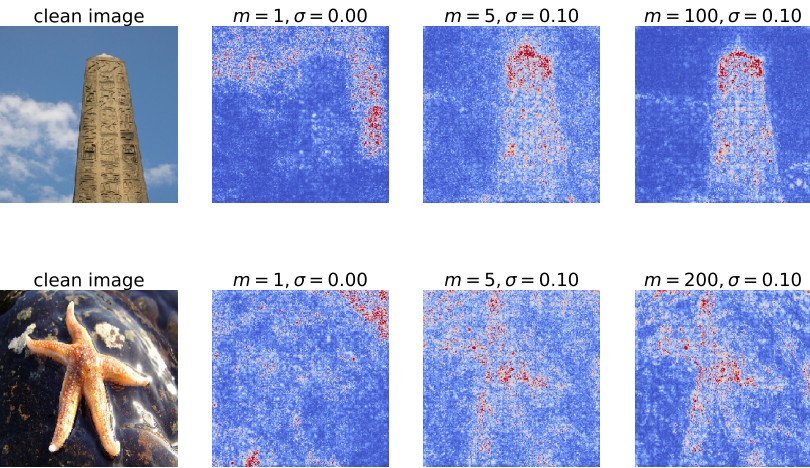

Figure 6: Visualization of gradients. The leftmost is the examined image. The second corresponds to the original gradient, whereas the remaining two images corresponds to the smoothed gradients estimated by different $m$.

## B  INFLUENCE OF STEP SIZE FOR TARGETED ATTACKS

When using IGSM to perform targeted black-box attacks, there are two hyper parameters including number of iteration $T$ and step size $\alpha$. Here we explore their influence to the quality of adversarial examples generated. The success rates are calculated on $5,000$ images randomly selected according to description of Section 4. *resnet152* and *vgg16_bn* are chosen as target models. The performance are evaluated by the average Top-5 success rate over the three ensembles used in Table 4.

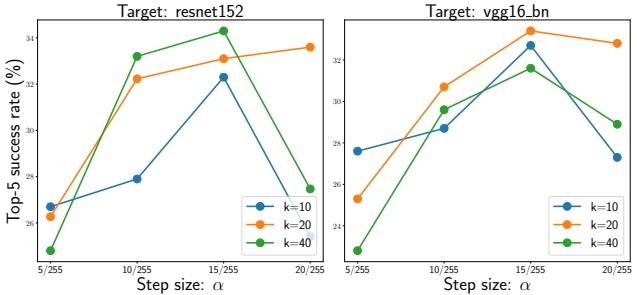

Figure 7: Average success rates over three ensembles for different step size $\alpha$ and number of iteration $k$. The three ensembles are the same with those in Table 4. Distortion $\varepsilon = 20$.

Figure 7 shows that for the optimal step size $\alpha$ is very large, for instance in this experiment it is about 15 compared to the allowed distortion $\varepsilon = 20$. Both too large and too small step size will yield harm to the performances. It is worth noting that with small step size $\alpha = 5$, the large number of iteration provides worse performance than small number of iteration. One possible explanation is that more iterations lead the optimizer to converge to a more overfitted solution. In contrast, a large

## C ADDITIONAL RESULTS

**How Model Capacity and Test Accuracy Affect Transferability** In this part, we additionally explore the impact of model capacity and test accuracy by using *resnet152* as our target model and perform transfer-based attacks against from a variety of models and the results shown in Figure 8. The observations are consistent with the observation in Section 5.2.

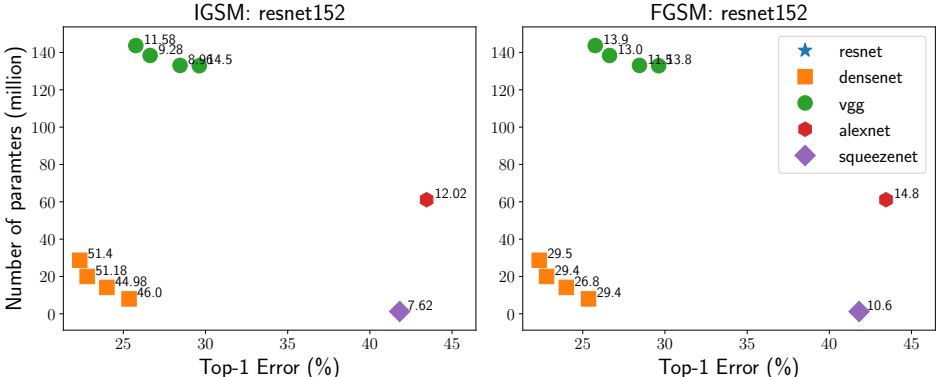

Figure 8: Top-1 success rates of IGSM ($T = 20, \alpha = 5, \varepsilon = 15$) and FGSM($\varepsilon = 15$) attacks against *resnet152* for various models. The annotated value is the success rate transferring to the *resnet152*. Here, the models of resnet-style have been removed to exclude the influence of architecture similarity. For the same color, the different points corresponds networks of different widths.

**Visualization of Decision Boundary** In this part, we provide additional results on the visualization of the decision boundary. Different Figure 3b, we here consider the sign of two directions, since the attack method actually updates using the sign $(g)$ rather than $g$. In Figure 9, we show the decision boundary. Each point corresponds to perturbed image:

$$\text{clip}(x + u\text{sign}(G_A) + v\text{sign}(h_A), 0, 255).$$

Here the model A is resnet34, and the definitions of $G_A$ and $h_A$ are the same as before. The color denotes the label predicted by the target model, and the gray color corresponds to the ground-truth label. We can see that the smoothed gradient $G_A$ is indeed more transferable than the noise part $h_A$.

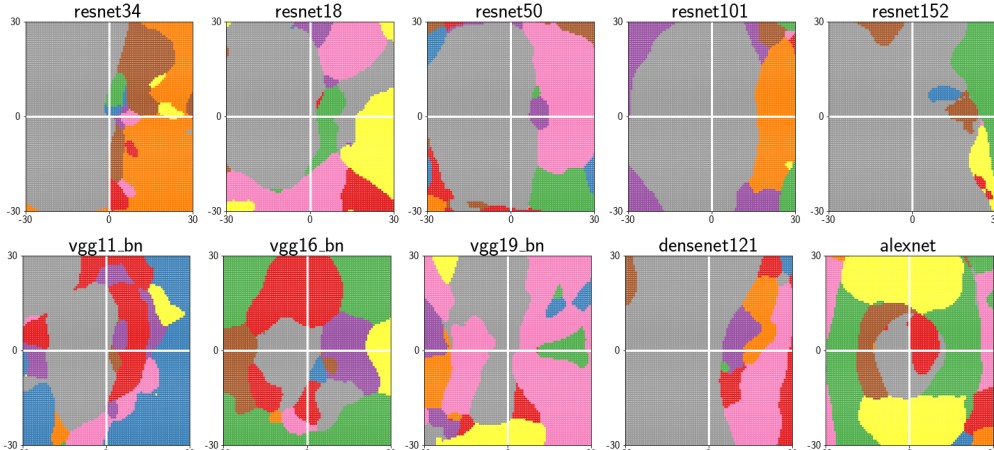

Figure 9: Visualization of decision boundaries. The source model is resnet34, and 9 target models are considered. The horizontal and vertical direction corresponds to $u$ and $v$, respectively.

**Additional Results for the Effectiveness of Smoothed Gradient Attacks**    In this part, we present some additional results to show the effectiveness of smoothed gradient attack.

Table 6: Top-1 success rates (%) of ensemble-based non-targeted IGSM and sg-IGSM attacks. The row and column denote the source and target model, respectively. The left is the success rate of IGSM ($T = 100, \alpha = 3$), while the right is that of sg-IGSM ($T = 50, \alpha = 3$). The distortion $\varepsilon = 20$.

| Ensemble | densenet121 | resnet152 | resnet50 | vgg13_bn |
|---|---|---|---|---|
| resnet18+resnet34+resnet101 | 87.8 / **97.8** | 94.6 / **98.9** | 97.4 / **99.4** | 84.1/96.1 |
| vgg11_bn+densenet161 | 86.8 / 97.2 | 62.9 / 89.7 | 80.3 / 94.8 | 94.9 / 98.4 |
| resnet34+vgg16_bn+alexnet | 68.9 / 91.3 | 54.6 / 87.2 | 77.9 / 96.2 | 98.1 / **99.1** |

Table 7: Top-1 success rates (%) of ensemble-based targeted IGSM and sgd-IGSM attacks. The row and column denote the source and target model, respectively. The left is the success rate of IGSM ($T = 20, \alpha = 15$), while the right is that of sg-IGSM ($T = 20, \alpha = 15$). The distortion $\varepsilon = 20$.

| Ensemble | resnet152 | resnet50 | vgg13_bn | vgg16_bn |
|---|---|---|---|---|
| resnet101+densenet121 | 11.6 / 37.1 | 11.9 / 34.5 | 2.6 / 10.5 | 2.6 / 14.1 |
| resnet18+resnet34+resnet101+densenet121 | 30.3 / **55.2** | 36.8 / **57.3** | 10.8 /**29.1** | 12.8/35.0 |
| vgg11_bn+vgg13_bn+resnet18+ resnet34+densenet121 | 10.1 / 35.1 | 22.2 / 47.9 | - | 42.1/**72.1** |

## D    THE EXAMINED IMAGE FOR VISUALIZATION OF DECISION BOUNDARY

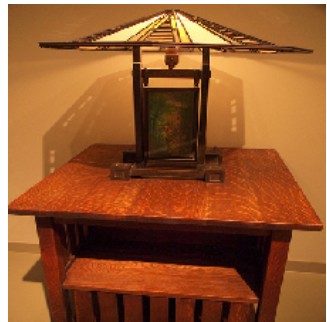

Figure 10: The image used to perform decision boundary visualization. Its ID in ILSVRC2012 validation set is 26, with ground truth label being *table lamp*.

