# OpenReview forum: "Exploring and Enhancing the Transferability of Adversarial Examples"
_ICLR.cc/2019/Conference_

### Official Review · AnonReviewer3 · 2018-10-26
**Enhancing the Transferability of Adversarial Examples with Smoothed Gradient Attacks**

**Rating:** 6
**Confidence:** 3

**Review:**

Summary. The authors empirically investigate the influence of the architecture and the capacity of an NN-model on the transferability of adversarial examples. They also study the influence of the smoothness. From the obtained results, they propose the smoothed gradient attack showing improvements on the transferability of adversarial examples.

Pros.
* Robustness of neural nets is a challenging problem of interest for ICLR
* The paper is well written
* The experimental study is convincing
* The experimental results for the smoothed gradient attacks are promising

Cons.
* The results of the experimental study are somehow expected
* the idea of smoothing gradients is not new

Evaluation.
The experimental study of the transferability of adversarial examples is well designed. Experimental protocol is convincing. The smoothed gradient attacks improve many previously proposed attacks. Therefore, my opinion is rather positive. But, as a non expert in the field, I am not completely convinced by the novelty of the approach.

Some details.
Typos: That l8 abstract; systems l9 intro; and l2 related work; directly evaluation l2 Section4, must has l-10 p4;
* the choice \sigma = 15 in Section 6.2 should be justified by the following study
* \sigma is not given in Figure 3(a)

---

> ### Author Response · Authors · 2018-11-26
> **Response to Reviewer 3**
>
> Thank you for approving our contribution to understanding the transferability of adversarial examples.
>
> We agree with that the smoothing gradient idea, especially the Gaussian smoothing technique, is not new, since the smoothing strategy could be used in many different scenarios. However, we motivate and derive the idea of smoothing the gradient based on our novel understanding on the transferability of adversarial examples between two models. To the best of knowledge, we are the first to derive and apply this technique to enhance the transferability of adversarial examples, whose significant improvement is also confirmed by our intensive experiments.

---

### Official Review · AnonReviewer2 · 2018-10-31
**Interesting insights and attack broadening our understanding of the scope of the problem of adversarial examples**

**Rating:** 6
**Confidence:** 3

**Review:**

The paper explores how the architecture, smoothness of the decision boundary and test accuracy of a model impacts the transferability of examples produced from it.  The paper provides a couple of novel insights, such as the asymmetry when transferring adversarial examples from one model to another. In addition, a novel method is proposed to enhance the transferability of adversarial examples from any model, through using smoothed gradients.

The experiments seem to show that the effect is rather large, and also makes the examples more robust to other transformations such as JPEG compression. Overall, these are interesting insights that could lead to further developments in making models more robust to adversarial examples. In particular, deriving adversarial examples that are both transferable and resilient to certain usual image transformations shows that the scope of the issue with adversarial examples may be even greater than what is understood today.

The paper is rather clear. Unfortunately, it is riddled with grammatical errors and should be proof-read carefully. A lot of singular/plurals are off, and some formulations are odd or downright unclear. Some examples (there are way too many to report them all):

- "Transfer-based attackS ... since they ...*
- "of adversarial exampleS ..."
- "from model A can transfer to model B"
- "less transferable than *those from* a shallow model"?
- "investigations, We ": don't capitalize
- "the averaging *has* a smoothing effect"
- "our motivation are"
- "contributed it to"
- "available *to the* adversary"
- "crafting adversarial perturbationS"
- "directly evaluation"
- "be fixed 100"

Pros:
- Transferability and robustness of adversarial examples is a very important problem
- Interesting insights, esp. the construction and evaluation of examples that are more resilient to certain image transformations
- Experimental results are convincing

Cons:
- Contribution overall may be a bit limited
- Grammatical errors and odd formulations all over the place

---

> ### Author Response · Authors · 2018-11-26
> **Response to Reviewer 2**
>
> Thank you for the appreciation on the novelty of our paper.
>
> - We have carefully proofread the manuscript and fixed the typos in the revised version. Could reviewer be more specific about the odd formulations, so that we can improve them?

---

### Official Review · AnonReviewer1 · 2018-11-01
**Paper that presents an experimental study of adversarial examples transferability with a contribution based on loss smoothing. The study is interesting with good illustrations and potential improvements but seems a bit limited, the conclusions are rather expected or unsurprising.**

**Rating:** 4
**Confidence:** 2

**Review:**

This paper addresses the problem of adversarial transferability, i.e. the ability that an adversarial example generated by one model can successfully fool another model. There are numerous papers on this topic recently, such as Fawzi'15, Liu'17, Dong'18, Athalye'18...
The authors propose tot study two types of factors that might influence transferability: model-specific parameters and smoothness of loss surface for constructing adversarial examples. Two experimental studies are made for each influence factor from existing architectures. Another attack strategy aiming at smoothing the loss surface is proposed, an experimental evaluation shows the effectiveness of the proposed method.

Pros
-the proposed experimental studies can be interesting to the community
-many interesting illustrations are provided.

Cons
-The conclusions of the study were suggested by previous papers or are rather expected: adversarial transfer is not symmetric: Deep models less transferable than shallow ones, averaging gradient is better
-I find the experimental studies a bit limited and I would expect larger studies which would have improve the interest of the paper.
-Only two influence factors are studied, again the paper would be more interesting with a more general study

The paper has an interesting potential but seems a bit limited in its present form.

---

> ### Author Response · Authors · 2018-11-26
> **Response to Reviewer 1**
>
> Thank you for the comments. We provide the feedbacks below.
>
> - Indeed there are a huge number of papers on adversarial examples, but specifically only a small fraction  of them are about the  transferability of adversarial examples . Understanding why adversarial examples can transfer from one model to another model is a much harder problem, which is a rather unexplored area. Indeed we do not provide a perfect explanation in this submission, however the factors we have considered and the well-designed numerical investigations could be helpful for future studies on this topic. In addition, the works Fawzi'15 and Athalye'18 did not talk about the issue of adversarial transferability.
>
> - Could you be more specific on what do you expect for "larger studies" and "general study”?  This will be helpful for improving our work.

---

> > ### Comment · AnonReviewer1 · 2018-12-02
> > **Response on adversarial transferability**
> >
> > Thanks for the answer.
> > Note that  the paper "Universal adversarial perturbations" by Moosavi-Dezfooli et al. (Oral CVPR 2017)
> > https://arxiv.org/pdf/1610.08401.pdf
> > The paper is rather popular (got more than 240 citations) and addresses also the problem of transferability and generalization, other related references citing this paper can be found.
> > This paper deserves to be cited and by larger and more general study (experimental study provided by authors is actually large), I would expect more discussion and comparison about existing work.

---

> > > ### Author Response · Authors · 2018-12-04
> > > **Moosavi-Dezfooli 2017 did not study the transferability of adversarial examples**
> > >
> > > Thanks for the reference, and we will add discussions about universal adversarial perturbations. However, in a nutshell, Moosavi-Dezfooli 2017 did not study the “transferability of adversarial examples”.
> > >
> > > Mossavi-Dezfooli 2017 showed the existence of a “universal” (image-agnostic) perturbation that causes most of the nature images to be misclassified. This is very different from the “transferability” addressed in our paper in two aspects.
> > >
> > > (1) The “transferability of adversarial examples” actually refers to the cross-model generalization, i.e. perturbations generated for one specific model can also fool the other unseen models. The “universal perturbations” studied in Mossavi-Dezfooli  2017, however, refers to the cross-image generalization, which means one perturbation can lead to misclassify most of the natural image simultaneously.
> > >
> > > (2) For adversarial transferability, we don’t need to generate adversarial examples that can fool a large number of models simultaneously. In practice, usually one model is enough to guarantee transferability.  Fooling a model ensemble can enhance the transferability, but it is not necessary.  In contrast, according to Figure 6 in Moosavi-Dezfooli 2017, at least thousands of training images  are required to guarantee the cross-image generalization, i.e. the universality talked in the paper. To some extent, the later can be understood in the framework of statistical learning theory, however the former can not be .
> > >
> > > In the experiments, Moosavi-Desfooli 2017 did demonstrate that the universal perturbations can transfer among different models. However they didn’t study the issue.

---

### Meta-Review · Area_Chair1 · 2018-12-17
**Needs improvements.**

**Confidence:** 4
**Recommendation:** Reject

**Metareview:**

While the paper contains significant information, most insights have already been revealed in previous work as noted by R1.
The empirical novelty is therefore limited and the authors do not provide theoretical analysis to complement this.